# Extremely Low-Frequency Electromagnetic Fields Increase Cytokines in Human Hair Follicles through Wnt/β-Catenin Signaling

**DOI:** 10.3390/biomedicines10040924

**Published:** 2022-04-18

**Authors:** Ju-Hye Choi, Yu-Mi Kim, Hee-Jung Park, Myeong-Hyun Nam, Young-Kwon Seo

**Affiliations:** Department of Medical Biotechnology, Dongguk University, Goyang-si 10326, Korea; jooh031919@gmail.com (J.-H.C.); kjmtik@nate.com (Y.-M.K.); gnflwldk98@naver.com (H.-J.P.); iis05047@naver.com (M.-H.N.)

**Keywords:** hair follicle, dermal papilla cells, hair bulb spheroids, electromagnetic field, β-catenin

## Abstract

Hair loss is a chronic disorder that affects many people; however, a complete treatment has not yet been developed. Therefore, new therapeutic agents for preventing hair loss must be developed, and electromagnetic field (EMF) therapy has been proven to be a promising medical treatment in various fields, including hair loss treatment. This study evaluated the effect of extremely low-frequency electromagnetic field (ELF-EMF) intensity and exposure time by analyzing the expression of cytokines and anagen-related molecules, which influence hair activation and growth, in hair bulb spheroid (HBS) and hair follicle (HF) organ cultures. ELF-EMFs did not induce toxicity in the HBSs, as verified via the lactate dehydrogenase (LDH) assay. Moreover, an ELF-EMF intensity of 5–20 G promoted the expression of ALP, versican, β-catenin, and several cytokines (VEGF, PDGF, FGF-10, and ET-1) in HBSs. Immunohistochemical staining showed that ELF-EMF at an intensity of 5–20 G upregulated ALP and β-catenin and decreased TUNEL staining in HBS. Moreover, HFs exposed to ELF-EMF for 60 min exhibited an increase in hair length and a 1.5-fold increase in IL-4, ICAM-1, ALP, and versican mRNA expression compared to the control. Immunohistochemical staining indicated that 60 min of ELF-EMF can increase the expression of ALP and β-catenin and decreases TUNEL staining in organ cultures. Collectively, our results demonstrated that ELF-EMF exposure at a 10 G intensity for 60 min promoted hair shaft growth in HFs due to the effect of cytokines and adhesion molecules via the Wnt/β-catenin pathway. Therefore, ELF-EMF is a promising treatment for hair loss.

## 1. Introduction

The development of novel therapeutic agents for hair loss prevention has recently garnered increasing attention. For example, minoxidil is a well-known and widely used treatment against hair loss. Minoxidil treatment activates β-catenin in human dermal papilla cells, thereby prolonging the anagen phase and delaying catagen progression in mice [1]. Similarly, valproic acid promotes β-catenin expression and elongates human hair shaft growth compared to control human hair organ cultures [2]. Caffeine added ex vivo to cultured human hair follicles (HFs) over 5 days enhanced hair shaft elongation by 96% compared to the control [3].

Recently, physical stimuli such as LEDs, lasers, and electrical stimulation have emerged as novel treatments for hair loss. Light-emitting diode (LED) irradiation and low-level laser therapy (LLLT) induce the proliferation of dermal papilla (DP) cells in vitro [4,5]. Human HFs irradiated with LED light for 3 days grew 20% longer than the controls [4]. Furthermore, HFs exposed to radiofrequency radiation for 60 min per day for 7 days grew 0.5 mm more than the controls [6]. Additionally, electrical stimulation increased the expression of hair-related genes in DP cells and rabbit skin [7].

Extremely low-frequency electromagnetic fields (ELF-EMF) have been reported to be effective against several medical conditions, including hair loss [8,9]. Recent reports demonstrated that a low-frequency electromagnetic field (5 mT, 50 Hz, 30 min/day) can induce hair growth [10,11]. Li et al., found that EMFs induce high keratin expression, suggesting that electromagnetic treatment (50 Hz, 50 G) can enhance hair growth in mice. These EMF-treated mice exhibited a more robust K15^+^ stem cell proliferation and increased expression of KGF [10]. When nude mice were treated with EMF after being injected with a mixture of epidermal stem cells and DP cells, the cell mixture formed high-density epidermal layers and hair bulbs in the dorsal skin [11]. However, these studies did not elucidate the mechanism for the effect of EMF. In a previous study, we reported that ELF-EMFs can activate the anagen-related molecules versican, ALP, laminin, and collagen IV, which were strongly upregulated in DP single-cell cultures by ELF-EMF exposure. Previous studies have suggested that ELF-EMF treatment promotes the activity of Wnt3α/β-catenin in human DPs, a signaling pathway induced by the phosphorylation of AKT and ERK, which inactivates GSK3β. In turn, this results in increased β-catenin accumulation and proliferation of DP cells in monolayer cultures [12]. Our findings demonstrated an increase in the expression of anagen-related molecules; therefore, we concluded that the activation of DP cells was induced by the GSK-3β/ERK/Akt signaling pathway. These results were confirmed through single-cell monolayer culture. However, given that actual hair is composed of various cells, assessing the effects of EMF using only a single cell line cannot provide insights into the cell–cell interactions and communication mechanisms that promote follicle growth. DP, outer root sheath (ORS), dermal sheath, and dermal matrix cells regulate the hair follicle activity and growth through their paracrine factors and cell–cell adhesion.

Few reports have evaluated the mechanisms of ELF-EMFs activation of hair growth using 3D culture and organ culture models. Therefore, this study evaluated the efficacy of ELF-EMF intensity and exposure time on hair cells and analyzed its effect on the expression of growth factors, adhesion molecules, and cytokines associated with hair activation in hair bulb spheroid (HBS) and hair follicle (HF) cultures. In summary, our study determined whether ELF-EMF affects hair follicle growth by assessing its effects on organ cultures.

## 2. Materials and Methods

### 2.1. Primary Culture of DP and ORS Cells

HFs were obtained from hair transplantation surgeries upon patient approval by the institutional review board (DUIRB approval No. 201904-11, 3 May 2019).

The HFs were dissected into single pieces from scalp skin and cut at the level of the sebaceous duct under a stereomicroscope at a 40× magnification (SAMWON, Goyang, Korea). Only anagen (anagen VI) HFs were used in this study, as previously described [13]. The DP and ORS cells were then separated from donated HFs. To separate the DP cells, HFs were cut into the bulb part, then treated with 3 U of dispase II (Sigma, St. Louis, MO, USA) in a 37 °C incubator for 1 h. The hair bulbs were then carefully opened, after which DPs were extracted and treated with 2 mg/mL of collagenase (Worthington Biochemical Corp., Lakewood, NJ, USA) at 4 °C for 30 min. The DPs were separated into single cells and collected using centrifugation at 1000 rpm for 10 min. The DP cells were then resuspended in medium and seeded into a 35-mm culture dish. ORS cells were isolated in the same way using whole follicles without the bulb and hair shaft.

DP cells were cultured with Dulbecco’s modified Eagle’s medium (DMEM) supplemented with 10% fetal bovine serum (FBS) containing antimycotic-antibiotic (Gibco-BRL, Rockville, MD, USA). ORS cells were cultured in a keratinocyte-serum free medium (K-SFM, Gibco-BRL, Rockville, MD, USA) supplemented with 5 mg/L of epidermal growth factor and 50 mg/L of bovine pituitary extract (Gibco-BRL, Rockville, MD, USA).

### 2.2. EMF Exposure

Figure 1 shows the pulsed ELF-EMFs device in a 37 °C, 5% CO_2_ incubator. We evaluated the effect of EMF intensity and exposure time on the HBSs and HFs, respectively (Table 1). The HBSs were treated with ELF-EMFs at varying intensities (2, 5, 10, 20, and 50 G) at a 60 Hz frequency for 15 min per day for 7 days. In another experiment, the HFs were treated with ELF-EMFs for 15, 60, and 180 min each day at a 60 Hz frequency over 14 days. The stimulus waveform was pulsed, with a 0.25 μs duration and a 10 G intensity. The control group was located in a separate incubator to avoid ELF-EMF exposure, and the culture media were identical in all experiments.

### 2.3. Hair Bulb Spheroid Culture

3D spheroid HBSs were formed using silicone elastomer-based concave microwells (StemFIT 3D; diameter of 600 μm and 389 wells, MicroFIT, Seongnam, Korea). The 1 × 10^5^ DP cells were seeded in each microwell and cultured with DMEM containing 10% FBS. After incubation for 1 day, 1 × 10^5^ of ORS cells were seeded with “FAD-media” with 10% FBS in each concave microwell containing the DP spheroids and incubated on a shaker at 40 rpm for 1 day to form the HBSs. The diameter of the final HBSs was approximately 100–150 μm, and they were maintained in DMEM at 37 °C and 5% CO_2_ for 7 days. All media were replaced every day.

### 2.4. Human Hair Follicle Organ Cultures

Only anagen phase (anagen VI) follicles with no visible microscopic damage were selected from the hair follicles isolated from the hair transplantation surgery patients. Individual and free-floating follicles were placed into 12-well cell culture dishes. Human HFs were incubated in DHGM at 37 °C and 5% CO_2_ for 14 days. All culture media were replaced every three days.

Hair shaft elongation was measured at day 7 and day 14 and HFs in the anagen phase were evaluated based on their growth. The HF length was evaluated as the entire length from the base of the hair bulb to the tip of the hair shaft on day 7 and day 14 after the experiment. After 14 days of culture, mRNA and protein analysis were performed for HFs, and an immunohistochemical staining assay was performed on three hair follicles.

### 2.5. Lactate Dehydrogenase Activity Assay

Cytotoxicity was evaluated via the lactate dehydrogenase assay (LDH-LQ kit; Asan Pharmaceutical co., Seoul, Korea). Briefly, 100 μL of the media collected from the HBS and HF cultures on days 7 and 14 were placed in a 96-well plate. Afterward, 50 μL of the working solution was added to a 96-well plate for 30 min at room temperature. Next, 1 N HCl (50 μL) was added and the absorbance was measured at 490 nm.

### 2.6. Quantitative Real-Time PCR

Total RNA was extracted from the HBSs and the HFs using 500 μL of TRIzol reagent (Invitrogen, Waltham, MA, USA) per sample. After sonication, 100 μL of chloroform (Sigma, St. Louis, MO, USA) was added to the samples, after which the solution was mixed and incubated for 3 min. The samples were then centrifuged at 14,000 rpm and 4 °C for 20 min. The supernatant was then transferred to a new tube and 300 µL of isopropanol was added. Next, the samples were incubated for 10 min and centrifuged once again for 15 min at 14,000 rpm, after which the supernatant was discarded. Next, 1 mL of 70% ethanol was added to the pellet, followed by centrifugation at 9000 rpm at 4 °C for 5 min. The supernatant was then discarded and the pellet was dried at room temperature. Next, 20 μL diethylpyrocarbonate (DEPC)-water was added to the samples and the solution was placed on ice. The total RNA was quantified using a Nanodrop spectrophotometer (Thermo Fisher Scientific, Waltham, MA, USA) and cDNA was synthesized using a RT master mix (Clontech, Palo Alto, CA, USA).

Gene-specific primer pairs for caspase-3, vascular endothelial growth factor (VEGF), platelet-derived growth factor (PDGF), fibroblast growth factor-10 (FGF-10), endothelin-1 (ET-1), versican (VCAN), intracellular adhesion molecule 1 (ICAM-1), alkaline phosphatase (ALP), interleukin-4 (IL-4), and β-actin were purchased from Bioneer (Bioneer, Daejeon, Korea). The sequences of these primers are described in Table 2. Quantitative real-time PCR (qPCR) was performed by StepOnePlus™ Real-Time PCR Systems (Applied Biosystems, Waltham, MA, USA) using TB Green^®^ Premix Ex Taq™ (Takara Bio, Kusatsu, Japan) and analyzed by the comparative C_T_ (ΔΔC_T_) method.

### 2.7. Western Blotting

To conduct Western blot analyses, the HBSs and HFs were first lysed with 200 μL 1 × RIPA buffer (Dyne Bio, Goyang, Korea) and sonicated. Afterward, the cell lysates were denatured at 100 °C for 5 min. The total protein content of the cell lysates was determined using the BCA protein assay, and the proteins in each sample (40 μg total protein) were separated via 10% SDS-polyacrylamide gel electrophoresis (SDS-PAGE). The separated proteins were then electro-transferred from the gel onto a nitrocellulose membrane (Millipore Co., Burlington, MA, USA). The membranes were blocked with 5% fat-free skim milk in Tris-buffered saline (TBS) containing 0.1% Tween 20 (TBS-T buffer) at room temperature for 1 h. The membrane was then washed three times with TBS-T and incubated overnight with primary antibodies; Laminin, ALP, versican, β-catenin, Wnt3α, and β-actin (Abcam, Cambridge, MA, USA) diluted in 10% bovine serum albumin.

The membranes were then incubated once again with secondary antibodies (Abcam, Cambridge, MA, USA) and washed extensively in TBS-T to remove any excess secondary antibodies. The blots were visualized with an enhanced chemiluminescence reagent (Thermo Fisher Scientific, Waltham, MA, USA) and photographed using a ChemiDoc XRS+ gel imaging system (Bio-Rad, Hercules, CA, USA). The Western blot assays were representative of at least three experiments, and the results were analyzed using the ImageJ software (National Institutes of Health, Bethesda, MD, USA).

### 2.8. Immunohistochemistry and Imaging Analysis

Immunohistochemistry analyses were performed on cultured HBSs and HFs fixed at 4 °C for 12 h using 4% paraformaldehyde. Endogenous peroxidase activity was blocked using 0.03% hydrogen peroxide. Specific monoclonal antibodies against ALP (diluted 1:50, ab75699, Abcam, Cambridge, MA, USA), β-catenin (1:200, ab22656, Abcam, Cambridge, MA, USA), and TUNEL (11684817910, Roche, Branford, CT, USA) were used in our analyses. Finally, a standard immunohistochemical procedure was conducted with anti-rabbit immunoglobulin using the avidin-biotin-peroxidase complex (ABC) method.

Imaging analysis of these results were conducted using QuPath software (GitHub, San Francisco, CA, USA). Immunohistochemistry results were presented by percentage of positive cells or overall intensity of expression. Each data set was obtained by extracting and analyzing three annotations of the same size from the image.

### 2.9. Statistical Analysis

Each experiment was performed in triplicate and all data are presented as mean ± standard errors. Differences between multiple groups were identified via one-way analysis of variance (ANOVA) with Tukey’s post hoc test. Data that were not normally distributed were analyzed using the non-parametric Kruskal–Wallis test. Differences were deemed statistically significant at a *p*-value < 0.05 (* *p* < 0.05, ** *p* < 0.01).

## 3. Results

### 3.1. Evaluation of EMF Cytotoxicity

To evaluate the effect of different ELF-EMFs on DP and ORS cells, DP cells were separated from human hair follicles and an HBS in vitro model composed of DP cells and ORS cells was generated (Figure 2A,B). Approximately 1.4 × 10^4^ DP cells were obtained per hair follicle. Following our protocol, the DP and ORS cells were seeded in silicone elastomer-based microwells to generate HBSs, and the cells aggregated to form spherical organoids. The diameter of the three-dimensional cultured DP cells was similar to that of intact DPs [14,15,16].

Various ELF-EMF intensities were screened to analyze cytotoxicity and apoptosis in DP and ORS cells. The HBSs were treated with various ELF-EMF intensities (2, 5, 10, 20, and 50 G) at 60 Hz for 15 min per day. Figure 2C shows the cytotoxic effects of ELF-EMF on HBSs. Neither of the ELF-EMF intensities tested increased lactate dehydrogenase secretion, and we therefore concluded that these intensities did not induce stress (**^†^**
*p* > 0.5). Caspase-3 expression, an apoptosis biomarker, was assessed via quantitative real-time PCR (qPCR) to evaluate cell death in HBSs. As with the results of the LDH assay, none of the ELF-EMF intensities increased the expression of caspase-3 (Figure 2D, **^†^**
*p* > 0.5). Therefore, none of the ELF-EMF intensities evaluated in this study induced cytotoxic effects or apoptosis in the HBS in vitro model.

### 3.2. Analysis of HBS Activity after EMF Treatment

Next, immunohistochemical staining, mRNA analysis, and Western blotting were conducted to assess the effects of EMF treatment on the HBSs. The HBS organoids were exposed to ELF-EMF for 7 days. Then, immunohistochemical staining of ALP and β-catenin was conducted to evaluate the expression of anagen-related markers. Additionally, TUNEL staining was conducted to measure cell apoptosis in the HBSs (Figure 3). ALP staining tended to increase depending on the intensity of ELF-EMF compared to the control. However, its expression decreased at ELF-EMF intensities of 50 G. β-catenin expression also increased at 5 and 10 G intensities and decreased slightly at 20 G. TUNEL staining demonstrated that HBS apoptosis decreased in the groups treated with 5, 10, and 20 G ELF-EMF compared to the control group, while the 2 and 50 G groups increased. Therefore, our findings demonstrated that ELF-EMF exposure at 5, 10, and 20 G can increase the expression of ALP and β-catenin while decreasing TUNEL staining in HBSs.

Quantitative real-time PCR (qPCR) and Western blot analyses were performed to evaluate the expression of anagen-related factors in ELF-EMF-treated HBSs. The expression levels of growth factors (VEGF, PDGF, and FGF-10), a DP-specific molecule (versican), an adhesion molecule (ICAM-1), and a cytokine (ET-1) were measured using qPCR (Figure 4). After exposure to ELF-EMF at 2, 5, 10, 20, and 50 G for 7 days, the mRNA expression levels of all genes increased compared to the control. Particularly, VEGF mRNA expression levels increased between 10 and 20-fold in 5, 10, and 20 G. Similarly, PDGF mRNA expression levels gradually increased by approximately 15-fold in the 5, 10, and 20 G treated HBSs compared to the control. Moreover, FGF-10 mRNA expression levels increased 5-fold in the 5, 10, and 20 G treatment groups. ET-1 mRNA expression levels also increased 6-fold in the 2, 10, and 20 G treatments, and ICAM-1 mRNA expression level increased 10-fold in the 10 and 20 G treatments compared with the control. Versican mRNA expression levels increased approximately 6-fold in the 2, 5, 10, and 20 G treatment. Thus, the mRNA expression levels of VEGF, PDGF, FGF-10, ET-1, ICAM-1, and versican were significantly increased at 10 G and 20 G. Our Western blotting analyses demonstrated that the 10 G treatment increased laminin protein expression by 2-fold and all ELF-EMF intensities (2, 5, 10, 20, and 50 G) increased the expression of ALP protein from 3- to 5-fold. The β-catenin protein expression levels increased 2-fold in the 5, 10, and 20 G groups, whereas the 50 G intensity decreased its expression. The versican protein expression increased 3-fold at ELF-EMF intensities ranging from 5 to 20 G. Overall, our findings indicated that the 10 G ELF-EMF intensity can likely induce dermal papilla and ORS activity (Appendix A).

### 3.3. Evaluation of EMF Efficiency for Hair Growth and Activity

Overall, ELF-EMF treatment at an intensity of 10 G and a frequency of 60 Hz had an outstanding effect. Specifically, this treatment effectively activated anagen-related molecules in vitro. Based on the results of the HBSs, we next sought to identify the effects of different ELF-EMF exposure times on hair growth ex vivo.

A total of 288 HFs from nine different individuals were treated with ELF-EMFs for 15, 60, and 180 min at a 60 Hz frequency and a 10 G intensity. Figure 5A shows the hair shaft growth of HFs exposed to ELF-EMFs for 14 days. Hair shaft growth was measured using a stereomicroscope (40× magnification, SAMWON, Goyang, Korea). The HFs exposed to ELF-EMF for 60 min grew approximately 1.4 times longer than the controls on days 7 and 14 (* *p* < 0.05, ***p* < 0.01, respectively) (Figure 5B). The HFs exposed to ELF-EMF for 60 min had the highest increase in hair length compared to other groups. The average hair lengths of HFs exposed to ELF-EMF for 15, 60, and 180 min were 2.471, 3.194, and 2.468 mm, respectively, whereas the control group only reached a 2.390 mm length by the end of the experiment.

After organ culture, LDH and apoptosis assays were performed to assess hair damage or stress on days 7 and 14 (Figure 5C). LDH is an oxidative enzyme associated with cell membrane or cytoplasm damage [17]. Our finding indicated that the LDH levels were similar between the groups of cells exposed with ELF-EMF for 15, 60, and 180 min and the control. This means that ELF-EMF exposure for 15, 60, and 180 min did not significantly damage the HFs. To investigate the effect of exposure time on apoptosis, the expression level of caspase-3 (an apoptosis marker) was assessed through qPCR analysis (Figure 5D). The mRNA expression of caspase-3 showed no significant differences compared with the control group. Therefore, ELF-EMF treatment had no observable cytotoxic effects at the evaluated intensities.

### 3.4. EMF Regulates Anagen-Related Molecules in HFs in an Exposure Time-Dependent Manner

To determine whether the changes in HF gene expression by ELF-EMF were dependent on exposure time, HFs were exposed to ELF-EMF for 15, 60, and 180 min. Afterward, qPCR experiments were performed using specific primers. HFs exposed to ELF-EMF for 15 min exhibited slight increases in ALP and ICAM-1 expression (Figure 6A). Additionally, HFs exposed to ELF-EMF for 60 min exhibited an upregulation of all anagen-related genes compared to the other groups. ALP, ICAM-1, IL-4, and versican mRNA expression increased by approximately 1.5-fold in the 60 min exposure group compared with the control group. However, the 180 min exposure group exhibited no changes in any of the quantified genes except for IL-4. These results show that the ELF-EMF treatment for 60 min possibly stimulates and prolongs the anagen phase by activating the hair cells.

Western blot analyses were conducted to characterize the expression of anagen-related proteins (ALP and versican) and signaling-related proteins (Wnt3α and β-catenin) in HFs. After exposure to ELF-EMFs for 14 days, the protein expression profiles of the HFs were quantified via Western blot analysis. Different ELF-EMF exposure times had significant effects on the protein expression of HFs (Figure 6B,C). HFs treated with ELF-EMFs for 60 min had the highest expression levels of anagen-related molecules except for ALP. ALP protein expression levels gradually increased in all ELF-EMF-exposed groups and reached a maximum in the 180 min exposure group. Versican protein expression levels significantly increased 2-fold in the 60 min group compared with the control. Furthermore, Wnt3α and β-catenin protein expression levels increased by 3.5-fold and 2-fold, respectively, in the 60 min group compared with the control (*p* < 0.05).

To visualize the changes in the expression of the ALP and β-catenin proteins ex vivo, immunohistochemical staining was conducted after 14 days of culture (Figure 7). After treatment with ELF-EMF for 15–180 min, ALP expression in human hair dermal papilla highly increased (yellow circle; *p* < 0.01). ALP is known to be responsible for establishing and maintaining the structure of hair follicles during the hair growth cycle and it is strongly expressed in DP during the anagen phase and upregulates the Wnt/β-catenin pathway [18].

High expression levels of β-catenin were observed in the outer root sheath cells after 14 days of ELF-EMF treatment for 60 min (red arrow; *p* < 0.01). β-catenin is known to be activated in dermal matrix cells and ORS cells during the anagen phase [19]. TUNEL staining was conducted to evaluate apoptosis [20], and our findings indicated that TUNEL signals were weaker in the EMF exposure groups compared to the control. These results highlight the importance of ELF-EMF treatment time for hair growth and follicular activation.

## 4. Discussion

Given that intact DP cells in vivo are influenced by interactions with various surrounding cells, we considered the need for further study conducted using human hair bulb organoids and hair follicles. In this study, we hypothesized that various ELF-EMF intensities and exposure times would regulate hair activation and growth in cultured HBSs and HFs. Therefore, our study sought to assess the effect of different ELF-EMF intensities (2, 5, 10, 20, and 50 G) on the activation of the HBS model, as well as the effects of different ELF-EMF exposure times (15, 60, 180 min) on the growth of HFs in an organ culture model. Particularly, our study determined whether these variations in ELF-EMF intensities and exposure times could regulate signaling pathways, growth factors, dermal papilla-specific molecules, adhesion molecules, and cytokines.

We first examined whether ELF-EMF exposure induced cytotoxicity or apoptosis in HBSs. The ELF-EMF treatment did not induce cell membrane damage, stress, or apoptosis in the HBSs, as confirmed via the lactate dehydrogenase release assay (Figure 2C) and quantification of caspase-3 mRNA expression (Figure 2D). Based on these results, we concluded that ELF-EMF exposure had no cytotoxic effects.

Next, we analyzed the effects of ELF-EMF treatment on the expression of anagen-related genes in HBSs and HFs. To determine the optimal ELF-EMF intensity for promoting anagen-related gene expression, the effects of different daily ELF-EMF intensities (60 Hz frequency, 15 min exposures) were evaluated using the HBS in vitro model. Upon exposing the HBSs to different ELF-EMF intensities (2, 5, 10, 20, and 50 G) for 7 days, the expressions of VEGF, PDGF, FGF-10, ET-1, ICAM-1, versican, laminin, ALP, and β-catenin increased in the 5–20 G groups, particularly in the 10 G group.

VEGF (vascular endothelial growth factor), an important hair cell growth factor, is strongly expressed in DP cells during the anagen phase, stimulates proliferation in cultured ORS cells [21], and plays a significant role in angiogenesis associated with the human hair growth cycle [22]. Additionally, it can induce proliferation and migration of DP cells under an autocrine mechanism by acting directly on dermal papilla cells or stimulating local vascularization [23]. When human DP cells were cultured with minoxidil, the mRNA expression level of VEGF increased approximately 4-fold compared to the controls, which indicates that minoxidil contributes to the maintenance of dermal papilla vascularization and stimulation of hair growth [24]. ELF-EMF treatment also increased VEGF expression, meaning that this treatment could also stimulate and induce the proliferation of DP and ORS cells in HBSs. PDGF (platelet-derived growth factor) plays a role in stimulating the proliferation of dermal mesenchymal cells that may contribute to the formation of dermal papilla [25]. Additionally, this growth factor can induce epithelial cell growth and angiogenesis through autocrine mechanisms [26]. Postnatal PDGF-null mice develop a thinner dermis, distorted hair follicles and dermal sheaths, smaller dermal papilla, and thinner hair compared with wild-type mice. Additionally, Kiso et al., demonstrated that a combination of PDGF (a ligand of platelet-derived growth factor receptor alpha) and FGF promoted the proliferation of murine DP cells and enhanced the expression of anagen-related molecules associated with follicle stimulation and growth [27]. Furthermore, FGF-10 (keratinocyte growth factor-2) was found in the dermal papilla, and its receptor, FGFR2IIIb, was identified in the neighboring outer root sheath. Recombinant human FGF-10 significantly stimulates human hair follicle cell proliferation in organ culture [28]. Kawano et al., reported that the FGF family was strongly expressed in hair, particularly in the inner root sheath cells and hair bulge region. Moreover, topical FGF injection induced an earlier anagen phase and prolonged the mature anagen phase by activating β-catenin and Shh expression in mice [29,30]. ET-1 is an important cytokine in the migration, proliferation, and adhesion of various cells, including hair cells. ET-1 mRNA was strongly expressed in early passages of cultured DP cells, but its expression decreased in later passages, which was attributed to the ability of DPC to induce hair follicle regeneration [31]. Our results demonstrated that these growth factors involved in the induction of hair follicles and activation of hair cells are highly expressed after exposure to ELF-EMF at 5–20 G, and therefore ELF-EMF treatment may prolong the anagen phase.

Laminin, ALP, and versican are important extracellular molecular markers within the hair bulb, which are specifically expressed in DPs during the anagen phase. Laminin regulates hair follicular stem cell activity, induces the anagen phase of the hair cycle in vivo, and has an important role in the induction of hair elongation in hair germs [32]. Several studies have demonstrated that laminin is secreted by the follicular epithelium into the basement membrane separating the follicular epithelium and DP, where it interacts with β1 integrin receptors on the DP cell surface, thereby activating the Wnt/β-catenin pathway [33]. Versican, a large chondroitin sulfate proteoglycan molecule, is implicated in the induction of hair morphogenesis, the initiation of hair regeneration, and the maintenance of hair growth in human hair follicles [34]. The expression of versican in androgenic alopecia is lower than in normal human hair follicles and previous studies have detected anagen-specific expression of versican in dermal papilla using immunofluorescence staining [35]. Specifically, DP versican expression was significantly lower in old HFs than in young HFs, and old HFs exhibited low expression levels of extracellular matrix molecules and reduced hair shaft diameters [36]. Strong ALP activity is displayed during the entire hair cycle in dermal papilla, and only during late anagen and early catagen in the outer root sheath [37]. Moreover, ALP activity is increased from the bulbar dermal sheath, a cap-like structure surrounding the base of the hair bulb, to the dermal papilla. Other studies demonstrated that topical application of valproic acid to mice induced hair growth and strong ALP expression in dermal papilla [38]. Our results indicated that ELF-EMF treatment at a 5–20 G intensity increased anagen-related molecules to prolong the anagen phase of DP and ORS cells in HBS.

Most follicle-related studies have been exclusively conducted using monolayer and single cells. Here, cell–cell interactions were examined using actual hair follicle organ cultures. Based on the results of our HBSs culture assays, we concluded that 10 G was the most effective ELF-EMF intensity at a 60 Hz frequency, after which we examined the effects of different ELF-EMFs exposure times at 10 G on hair growth and the expression of anagen-related markers in HFs. The HFs exposed to ELF-EMF for 60 min grew the most. This ELF-EMF-induced hair shaft growth was likely due to the activation of anagen-related proteins and cytokines. Therefore, we next analyzed the expression of anagen-related markers in HFs.

ELF-EMF increased the expression levels of IL-4, ICAM-1, ALP, versican, and Wnt3α/β-catenin. IL-4 might be closely associated with alopecia [39]. Specifically, this gene can accelerate the induction of the telogen-to-anagen transition in vivo. Many studies have reported that IL-4 directly induces the telogen-to-anagen transition in the hair growth cycle and reductions in IL-4 have been linked to T-lymphocytes attacks on hair follicles. Other studies have demonstrated that IL-4 may promote hair growth by enhancing ASC migration [40]. Moreover, IL-4 treatment increased the length of mouse hair in organ cultures and proliferation of DP cells in vitro, in addition to inhibiting apoptosis in cultured human DP cells [41].

Our study also analyzed ICAM-1 expression to evaluate cell-to-cell adhesion. ICAM-1 transcription levels constantly change during the hair cycle. Particularly, its levels declined after depilation and peaked in the middle of the anagen phase. Other studies have reported that the catagen phase occurs quicker in ICAM-1 knockout mice compared to wild-type controls [42]. Another report demonstrated that VEGF stimulated ICAM-1 activation [43], and our study demonstrated that ELF-EMF can increase the mRNA expression of VEGF.

DP cells regulate the development of hair follicles in humans, and the Wnt/β-catenin pathway is considered essential for maintaining the hair induction activity of DP cells. β-catenin activity has been shown to maintain the characteristics of the growth phase in cultured DP cells [5]. The Wnt/β-catenin pathway plays an important role in hair follicle morphogenesis, proliferation, differentiation of hair cells, and the regulation of hair follicle recycling [12]. Our results demonstrated that the expression of Wnt3α and β-catenin increased after ELF-EMF exposure for 60 min. However, the expression of other important biomarkers of hair growth (ALP, ICAM-1, IL-4, and versican) also increased in response to ELF-EMF treatment.

In our previous study, we demonstrated that ELF-EMF treatment can induce the Wnt3α/β-catenin pathway in a monolayer DP cell culture model [12]. In this study, the HBS in vitro model and HFs organ culture model were used to confirm the expression of various growth factors. In the HBSs, the expression of anagen-related markers and hair cell proliferation markers reached a maximum at an ELF-EMF intensity of 10 G. In HFs, hair growth and the expression of these markers reached a maximum when the HFs were treated with ELF-EMF for 60 min. Thus, our findings suggest that ELF-EMF treatment at 10 G and 60 Hz for 60 min can increase hair shaft growth in HFs, which may be due to the induction of growth cytokines and adhesion molecules.

## 5. Conclusions

Very few studies have evaluated the potential applicability of ELF-EMF as a therapy for hair loss. Furthermore, to the best of our knowledge, no previous studies have assessed the effects of ELF-EMF exposure time on hair growth. Our study demonstrated that both ELF-EMF intensity and exposure time can affect the expression of anagen-related molecules and cytokines through the activation of the Wnt3α/β-catenin pathway in cultured human HFs. The roles of the Wnt3α signaling pathway in cell survival and death have been well established and Wnt3α is known to play a critical role in mediating survival signals. The activation of the Wnt3α pathway by ELF-EMF may be involved in regulating hair cell survival. Collectively, our findings suggest that ELF-EMF treatment can promote hair growth in HFs by activating dermal matrix cells, outer root sheath cells, and dermal papilla.

## Figures and Tables

**Figure 1 biomedicines-10-00924-f001:**
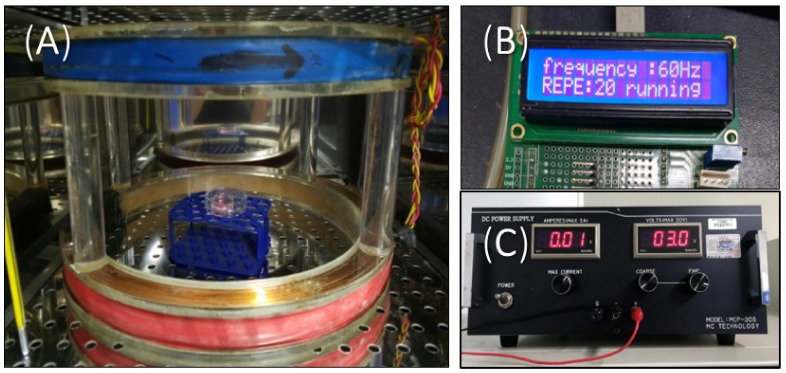
Photograph of the ELF-EMF device in a 5% CO_2_ incubator at 37 °C. ELF-EMF was generated using a pair of Helmholtz coils. (**A**) Helmholtz coil; (**B**) function generator; (**C**) power supply.

**Figure 2 biomedicines-10-00924-f002:**
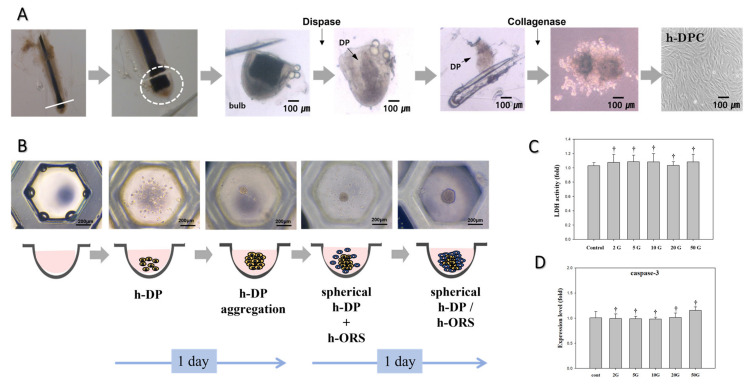
(**A**) Isolation of DP cells from hair follicles. (**B**) Establishment of the hair bulb spheroid (HBS) in vitro model consisting of human DP and ORS cells. (**C**) Evaluation of the cytotoxic effects of various ELF-EMF intensities (2, 5, 10, 20, and 50 G) on HBSs. Cytotoxicity was measured using the LDH assay kit. (**D**) Effects of various ELF-EMF intensities (2, 5, 10, 20, and 50 G) on HBS apoptosis. Apoptosis was measured via qPCR analysis of caspase-3. Each bar represents the mean ± standard error of independent experiments performed in triplicate (*n* = 3). Significant differences were determined via one-way ANOVA with Tukey’s post hoc test; ^†^
*p* > 0.5 compared to the control. Original magnification: 100×. Scale bar: 200 μm.

**Figure 3 biomedicines-10-00924-f003:**
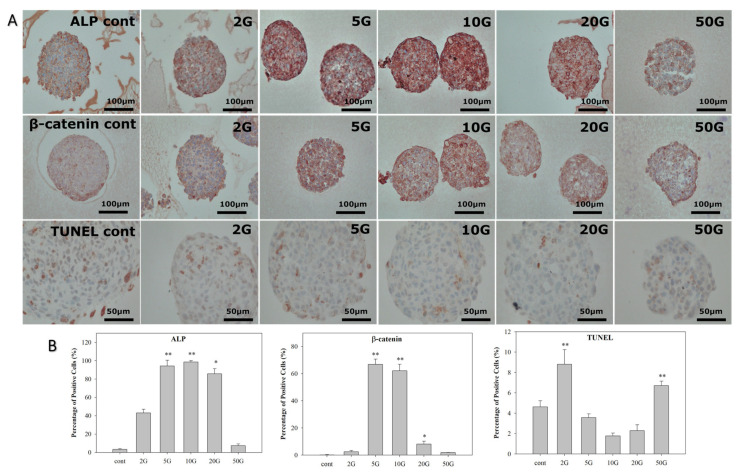
Immunohistochemical staining of ALP, β-catenin, and TUNEL assay in the hair bulb spheroid (HBS) in vitro model after simulation with ELF-EMF. (**A**) Scale bar: 100 μm. The brown color indicates positive staining. Original magnification: 200×; Scale bars: 100 μm and 50 μm. (**B**) Percentage of positive cells in (**A**). The values were calculated using QuPath software. Each bar represents the mean ± standard error of independent experiments performed in triplicate (*n* = 3). Significant differences were determined via the non-parametric Kruskal–Wallis test; * *p* < 0.05, ** *p* < 0.01.

**Figure 4 biomedicines-10-00924-f004:**
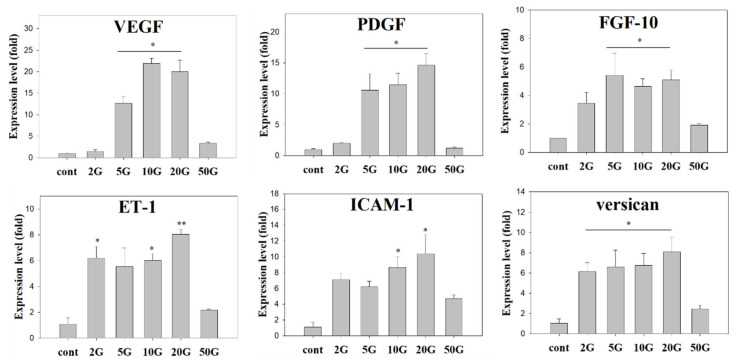
Gene and protein expression levels detected in hair bulb spheroids (HBSs) treated with ELF-EMF. Gene expression of VEGF, PDGF, FGF-10, ET-1, ICAM, and versican measured by qPCR using β-actin as an internal control. Each bar represents the mean ± standard error of independent experiments performed in triplicate (*n* = 3). Significant differences were determined via the non-parametric Kruskal–Wallis test; * *p* < 0.05, ** *p* < 0.01.

**Figure 5 biomedicines-10-00924-f005:**
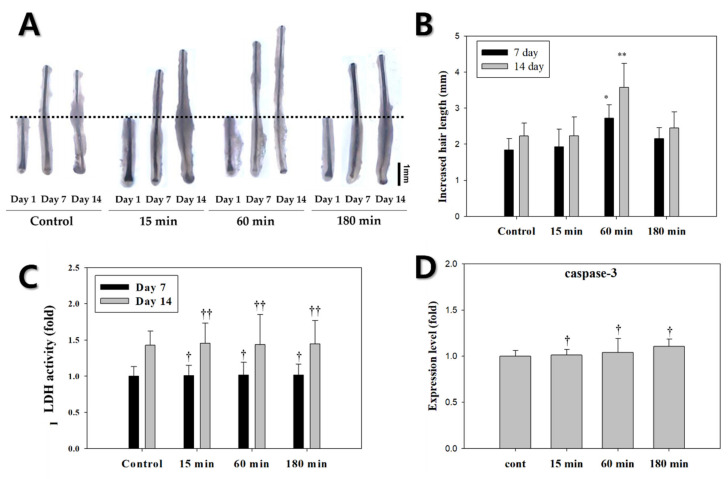
Effects of various ELF-EMF exposure times (15, 60, and 180 min) on human hair follicles (HFs). (**A**) Photograph of increased hair length on days 7 and 14. Original magnification: 40×; scale bar: 1 mm; (**B**) Increased hair length of human HFs (*n* = 6, female patient samples). (**C**) Evaluation of the potential cytotoxic effect of different ELF-EMF exposure times on HFs. Cytotoxicity was measured using an LDH assay kit. (**D**) Effect of the exposure times of ELF-EMFs on HF apoptosis. Apoptosis was measured via qPCR analysis of caspase-3. Each bar represents the mean ± standard error of independent experiments performed in triplicate. Significant differences were determined via one-way ANOVA coupled with Tukey’s post hoc test (* *p* < 0.05, ** *p* < 0.01). ^†^
*p*, ^††^
*p* > 0.5 compared to each control. The results are representative of at least six hair follicles per condition.

**Figure 6 biomedicines-10-00924-f006:**
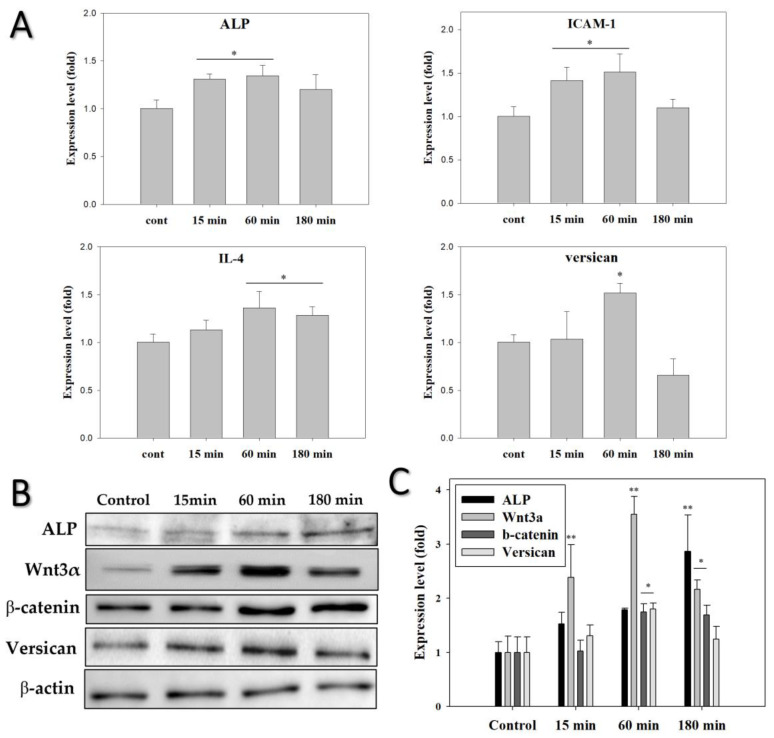
Gene and protein expression levels detected in human hair follicles (HFs) treated with ELF-EMF. (**A**) Gene expression of ALP, ICAM-1, IL-4, and versican measured by qPCR using β-actin as an internal control. (**B**) Protein expression of β-catenin, Wnt3α, ALP, and versican, using β-actin as an internal control. (**C**) Protein expression of (**B**). Relative expression intensities were calculated using ImageJ. Each bar represents the mean ± standard error of independent experiments performed in triplicate (*n* = 3). Significant differences were determined by one-way ANOVA with Tukey’s post hoc test (* *p* < 0.05, ** *p* < 0.01).

**Figure 7 biomedicines-10-00924-f007:**
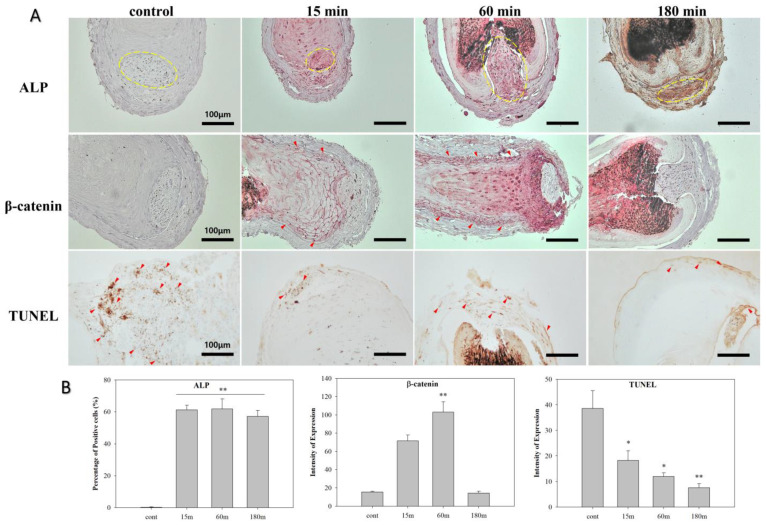
Immunohistochemical staining of ALP, β-catenin, and TUNEL in the hair follicles (HFs) after treatment with ELF-EMF for 14 days. Scale bar: 100μm. The arrows indicate positive staining. The dotted lines indicate dermal papilla. Original magnification: 200×. Scale bar: 100 μm. (**B**) Percentage of positive cells (ALP) or intensity of expression (β-catenin and TUNEL) in (**A**). The values were calculated using QuPath software. Each bar represents the mean ± standard error of independent experiments performed in triplicate (*n* = 3). Significant differences were determined via the non-parametric Kruskal–Wallis test; * *p* < 0.05, ** *p* < 0.01.

**Table 1 biomedicines-10-00924-t001:** Summary of the experimental conditions.

Model Tissue	ELF-EMF Irradiation Conditions	Irradiation Period
Hair bulb spheroid	Intensity: **2, 5, 10, 20, 50 G** Frequency: 60 Hz Exposure time: 15 min/day	7 days
Hair follicle	Intensity: 10 G Frequency: 60 Hz Exposure time: **15, 60, 180 min/day**	14 days

**Table 2 biomedicines-10-00924-t002:** PCR primer sequences. ET-1, Endothelin-1; FGF-10, Fibroblast growth factor 10; ICAM-1, Intercellular adhesion molecule 1; PDGF, Platelet-derived growth factor; VEGF, Vascular endothelial growth factor.

Gene	Primer Sequences	Amplicon Length (bp)
β-actin	5′-GAAAGCCTGCCGGTGACTAA-3′ 5′-TTCCCGTTCTCAGCCTTGAC-3′	301
Caspase-3	5′-TACCAGTGGAGGCCGACTTC-3′ 5′-GCGACTGGATGAACCAGGAG-3′	96
ET-1	5′-GCTGCCTTTTCTCCCCGTTA-3′ 5′-GCTTCAGGTCCCTCAAAGCG-3′	89
FGF-10	5′-AGCCCCAAACAACAACAACAG-3′ 5′-GCCATCCTCGTTTCCAATTCAT-3′	187
ICAM-1	5′-ACCATCTACAGCTTTCCGGC-3′ 5′-CAATCCCTCTCGTCCAGTCG-3′	293
PDGF	5′-GATCCGCTCCTTTGATGATC-3′ 5′-GTCTCACACTTGCATGCCAG-3′	435
VEGF	5′-GCCATCCAATCGAGACCCTG-3′ 5′-ATTAGACAGCAGCGGGCAC-3′	367
Versican	5′-GTGTTCCACCTCACTGTCCC-3′ 5′-CCTGGAGTTCCCCCACTGTT-3′	100

## Data Availability

The data generated and analyzed during this study are available from the corresponding author on reasonable request.

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
