# Peer review of "Extremely Low-Frequency Electromagnetic Fields Increase Cytokines in Human Hair Follicles through Wnt/β-Catenin Signaling"

_biomedicines, 2022, doi:10.3390/biomedicines10040924_

Round 1

Reviewer 1 Report

In their paper Choi et al. describe the effects of extremely low-frequency electromagnetic field (ELF-EMF) on cultured hair bulb organoids and hair follicles. The authors showed that ELF-EMF doesn’t affect viability of the hair follicle cells and stimulates expression of factors known to stimulate hair growth. They claim that ELF-EMF treatment stimulated WNT/b-Catenin pathway and may be thus a promising therapy for hair loss.

The study is not original as the authors duplicate some of their previously published observations. Furthermore, their statement about a role of the WNT/b-Catenin pathway is not supported by the present results.

The study suffers also from significant methodological flaws.

  1. The results of mRNA gene expression are not convincing. The authors evaluated specific mRNA expression by classical RT PCR method that is considered poorly quantitative. The authors should use a state-of-the-art quantitative real time PCR method.
  2. The same concerns the protein evaluations. Western-blot is also not a quantitative method and may be rather used just for confirmation of e.g. mRNA results.
  3. It is unclear whether increased expression ALP and b-Catenin (Fig. 3 and 7) is related to an increased number of positive cells, or it may also reflect an overall increased expression of these factors. Accordingly, the authors should enumerate positive cells and the intensity of reaction using some imaging method.
  4. From what is seen on the graphs the results of RT-PCR and WB analysis are not normally distributed (SD values exceed 50% of the means). Therefore, the authors should use non-parametrical statistical analysis.

Minor comments.

  1. The authors should decide whether they use black and white or color graphs.
  2. It would be of interest for other researchers to describe isolation and culture of follicle cells in more detailed way. What is missing is e.g., a method of cell isolation, cell yield etc.
  3. Discussion is poor as it duplicates the result description. It should be improved.
  4. I found few grammatical and syntax errors so it would be good to send the paper for English edition.

Author Response

Thanks for your time. Please see the attached file.

Reviewer 2 Report

Dear authors,

The manuscript “Extremely Low-Frequency Electromagnetic Fields Increases Cytokines in Human Hair Follicles through Wnt/β-catenin Signaling” is very interesting, well organised and presented. It fits perfectly with some of the latest research trends of biomedicine, providing several hints in this research area. Therefore, I strongly recommended the publication of this manuscript, after “minor revisions”.  Most of my revisions are reported and highlighted in yellow in the manuscript pdf version (please see the attached file). Specifically, you will find expressions crossed out in red with the relative correction and some reported as comments in the yellow box. Obviously, my corrections are only suggestions that I have proposed and that seemed more fitting to me while I was reading the manuscript. Furthermore, below there are some of my other comments/suggestions:

  • Materials and Methods: please, could the authors include a Table summarising the experimental conditions investigated?
  • Results: please, could the author integrate the statistics in the plots B and D in Figure 4 (Page 7)?
  • Have the authors already though a possible pre-clinical study in animal or even a possible clinical study aimed at validating the efficacy of this approach, or are they still far from this possible future perspective?

Author Response

(The authors gave the same response as above.)

Reviewer 3 Report

The paper by Choi et al. investigates the effects of extremely low-frequency electromagnetic fields on Human Hair Follicles through Wnt/ β-catenin signaling.

The conclusions are well supported by results obtained by the use of two suitable models: hair bulb organoids and hair follicles organ cultures.

I have 3 concerns:

  1. why do the authors use the term organoids and not spheroids?
  2. please revise the statistical analysis by the use of non-parametric tests: this should be done due to the large use of data normalization that provide non normally distributed data
  3. the text should be extensively revised for english language and style

Author Response

(The authors gave the same response as above.)

Reviewer 4 Report

The manuscript by Ju-Hye Choi described the effect of extremely low-frequency electromagnetic field (ELF-EMF) intensity on hair activation and growth with hair bulb organoid (HBO) and hair follicles (HF) organ culture. 

The study is interesting, but it presents several flaws that need to be addressed.

Major points

-The authors have to improve the results sections. In particular, the gene expression must be performed by using the real-time method; the apoptotic cells must be quantified; the statistical analysis introduced in the figures.

Minor points

The introduction, description of methods, discussion sections, must be improved.

The figures must e revised. In some cases, the labels are missing.

Author Response

(The authors gave the same response as above.)

Round 2

Reviewer 1 Report

The authors extensively corrected the manuscript in line with my previous criticism. However, there are still things that need to be addressed.

  1. The authors showed the results of qRT-PCR as requested. Accordingly, there is no longer needed to show the results of classical RT-PCR as they appear to be useless. Description of classical RT-PCR method and primer sequences should be removed, as well.
  2. The authors should add specific description of qRT-PRC method (syber green? Taqman?) and clearly indicate which method (delta? double delta?) was used for evaluation of specific mRNA expression.
  3. Information about a statistical method should be included in each figure caption where necessary.
  4. If ImageJ is not suitable the authors may use e.g., QuPath software for analysis of immunohistochemical reactions e.g., positive cell counts.

Reviewer 4 Report

no comments

Author Response

Thank you for your time.

Round 3

Reviewer 1 Report

The authors responded to all queries.

Author Response

Thank you for your time.